# Predictors of underweight among adult patients receiving antiretroviral therapy in Bishoftu general hospital, central Ethiopia: Case-control study

**Mesfin Gashaw Assefa**[1☯]**, Alem Deksisa**[2]**, Mariama Abdo**[2]**, Obse Tamiru Alemayehu**[1]**, Dawit Wolde Daka**[3☯]*

1 Bishoftu General Hospital, Bishoftu, Ethiopia, 2 Department of Public Health, Adama Comprehensive Specialized Hospital Medical College, Adama, Ethiopia, 3 Department of Health Policy and Management, Faculty of Public Health, Jimma University, Jimma, Ethiopia

☯ These authors contributed equally to this work.
* dawit.daka86520@gmail.com

## Abstract

### Background

Underweight affects the overall clinical outcome and quality of life and increases the risk of mortalities in Human immunodeficiency virus and acquired immunodeficiency syndrome (HIV/AIDS) patients. Though studies have examined the various determinants of being underweight in people living with HIV/AIDS (PLHIV), scanty evidence exists about the influence of dietary diversity scores and dietary counseling on underweight HIV patients in Ethiopia. This study aimed to identify the determinants of being underweight among adult patients receiving antiretroviral therapy at Bishoftu General Hospital, central Ethiopia.

### Methods

An institution-based unmatched case-control study was conducted among 279 participants (93 cases and 186 controls) from April to May 2022. Cases were selected consecutively as they occur, and then two subsequent controls that visited the antiretroviral therapy(ART) clinic were interviewed until the sample size was attained. Data were collected using a pre-tested structured questionnaire and involved patient interviews and chart review. Bivariate and Multivariable logistic regression was used to identify determinants of underweight. The presence of statistically significant association was declared with p-value <0.05, and a 95% confidence interval was used to show the precision in the measure of the strength of association.

### Results

The response rate of participants was 91.2% for each of the cases and controls. Monthly income of patients ≤2000 birr (AOR = 6.63, 95% CI: 2.96–14.85), absence of support giver (AOR = 3.22, 95% CI: 1.38–7.50), being having an eating problem (AOR = 14.48, 95% CI:

---

**Data Availability Statement:** All relevant data are within the paper and its Supporting information files.

**Funding:** The author(s) received no specific funding for this work.

**Competing interests:** The authors have declared that no competing interests exist.

5.06–41.40), dietary diversity score of four to five (AOR = 2.36, 95% CI: 1.92, 6.08), not getting dietary counseling support and advice (AOR = 2.53, 95% CI: 1.11, 5.72) and chewing khat (AOR = 3.36, 95% CI: 1.99,11.33) were determinants of underweight in adult HIV patients.

## Conclusions

This study revealed that household dietary diversity, counseling and education on nutrition, monthly income, eating problems, support giver, and khat chewing were predictors of being underweight in HIV patients. This inquires an integrated nutritional intervention including income-generating activities, counseling and education on nutrition and bad habits, and regular monitoring of the nutritional status during clinic visits.

## Introduction

Malnutrition, a nutritional deficiency, excess, or imbalance in individuals' intake of energy or nutrients, is interlinked with HIV/AIDS. Malnutrition reduces the capacity of the body to fight HIV infection by compromising the immune system. HIV/AIDS resulted in undernutrition through opportunistic infections by causing immune dysfunction in manifold ways [1, 2].

Underweight in adults is when BMI is less than 18.5 kg/m$^2$ [3]. HIV patients with underweight manifest various signs and symptoms that include moderate to severe weight loss, muscle wasting, chronic diarrhea for longer than one month, and anemia. A weight loss of about 6 to 7 kilograms in an average adult HIV patient is a sign of the onset of clinical AIDS. Therefore, it is recommended to maintain body weight in asymptomatic HIV-infected adults by 10% over acceptable levels as compared to healthy people [4, 5].

Undernutrition and HIV/AIDS are highly prevalent in sub-Saharan Africa and Ethiopia inclusive as linked in a vicious cycle. Several studies have documented that being underweight among HIV/AIDS patients increases the risk of mortality, and affects the overall clinical outcome and quality of life [6, 7].

Being underweight increases the risk of poor clinical outcomes and it is the predictor of the occurrence of opportunistic infections(OIs) among people living with HIV/AIDS [8]. Underweight significantly shortens the time to develop OIs in adults living with HIV compared to a well-nourished patient. The average time to develop OIs in underweight patients is estimated as 17 months and that of well-nourished is 26 months [9]. Underweight is also a predictor of lost-to-follow-up from ART services. The increased lost-to-follow-up impedes the progress towards the 2030 UNAIDS targets to end new HIV infections through high-impact prevention, accelerated HIV testing, treatment, and retention in care [10, 11].

Multiple factors are related to being underweight in HIV/AIDS patients. These include the level of education, marital status, family size, advanced disease stage, presence of opportunistic infections, CD4 count less than 200 cells/mm$^3$, substance use, eating problems, ART drug adherence, and the presence of nutritional counseling. Underweight is more common among HIV-positive adults with advanced disease stages, anemia, diarrhea, serum albumin < 3.5mg/dl, CD4 count of less than 200 cells/mm$^3$, eating problems, drug non-adherence, and smokers [7, 8, 12, 13].

Nutritional support is becoming an integral part of the ART program in African countries including Ethiopia. Nutritional counseling and support help that HIV patients maintain a healthy diet, manage illness, and monitor nutritional status. Eating well and a variety of foods

help to have a normal Body Mass Index (BMI), suppress illness, and maintain a healthy life in patients [14–16].

A comprehensive understanding of underweight determinants in HIV/AIDS patients has paramount importance programmatically. Though studies have examined the prevalence of underweight and determinants in HIV/AIDS patients [17–20], there is limited evidence on the influence of dietary diversity score and dietary counseling on underweight in Ethiopia. Therefore, this study aimed to examine the determinants of underweight in adult HIV/AIDS patients who were on treatment in Bishoftu general hospital, one of the high HIV spot sites in Ethiopia. The evidence will be used by the providers, program managers and policymakers to improve the care and treatment services provided to patients.

## Methods and materials

### Study design

The study has employed an institution-based unmatched case-control study design.

### Study setting and period

The study was conducted among adult HIV/AIDS patients on care and treatment in Bishoftu general hospital, central Ethiopia. Bishoftu town is one of the town administrations in the Oromia region located 40 kilometers in the Eastern direction of Addis Ababa. The town is a hub for culture and tourism, and the largest industrial zone of the country (Eastern Industrial Zone). The total population of the town administration was 238, 248 with male population accounted of 49% and that of female population was 51%. The town was one of the highest HIV spot areas in the country with a prevalence of 3% [21].

In the town administration, there were 6 health centers, one public hospital, one military hospital, and one private hospital. The public and military hospitals and two health centers were providing comprehensive HIV care and treatment services. Bishoftu general hospital provides services to more than 6000 pre-ART and ART clients. The patient flow rate in the ART clinic was 850 per month and approximately 30 patients per day. Currently, there were 5020 patients in the clinic, out of which 3874 were currently on ART. Of the registered patients, 82 were pregnant, 230 were children less than 15 years old, 928 were drop-out/defaulters, 111 have died, and 25 were lost-to-follow-up or hadn't been in contact with the clinic for more than three months. The remaining 3454 patients were active and receiving treatments. The study was conducted from April to May 2022.

### Study participants

The study participants were adult HIV/AIDS patients who were on treatment. Cases were all adult patients aged 15 years old and greater with BMI less than 18.5 $Kg/m^2$, and controls were all adult patients aged 15 years old and greater with BMI 18.5 to 24.9 $Kg/m^2$. All patients who were registered in Bishoftu general hospital, patients who had complete information on relevant variables, and those who visited the hospital during the data collection period were included. Patients who were pregnant, had mental health problems and critical illnesses, had less than 6 months' duration on ART, and who had been transferred from other health facilities were excluded.

The sample size was computed using Epi-Info (version 7.2.4) for an unmatched case-control study. The calculation was made based on several predictors of underweight those identified from previously conducted studies, and with the consideration of 95% confidence level, 80% power, and a case-to-control ratio of 1:2. The largest sample size was obtained using the

proportion of Adherence to ART (28.11% for cases and 12.6% for controls) and an Adjusted Odds Ratio (AOR) of 2.61 [22]. This yielded a total sample size of 279. After considering a non-response rate of 10%, the final calculated sample size was 306 (102 cases and 204 controls). Cases were selected consecutively as they occur, and then two subsequent controls that visited the ART clinic were interviewed until the sample size was attained. A cumulative density or survivor sampling strategy was used to select controls.

## Data collection

A structured questionnaire with a component of anthropometric measurement and dietary assessment was adapted from relevant literature, and modified to local context and research objectives [7, 11, 13, 23–25]. The questionnaire captures information such as basic socio-demographic variables, clinical characteristics, dietary diversity, substance abuse, and dietary counseling. A tool for measuring dietary diversity was adopted from the Food and Agricultural Organization (FAO) guideline for measuring individual dietary diversity [23]. Substance abuse including cigarette smoking, Khat chewing, and alcohol consumption was assessed using a structured questionnaire adapted from a STEPS survey on Non-Communicable Diseases (NCDs) risk factors in Ethiopia and the WHO STEP-wise approach to chronic disease risk factor surveillance [26]. Nutrition counseling was measured by using World Food Program (WFP) 2014 Guide on Nutrition Assessment, Counseling and Support for adults living with HIV from the specific needs [25]. The questionnaire was prepared in English and translated into the local language (Afan Oromo and Amharic). Then it was deployed on Open Data Kit (ODK) for data collection.

Data collectors were health professionals with a qualification of bachelor of science degree in nursing and the supervisor was a public health officer with a qualification of bachelor of science degree. Data collectors and supervisors were recruited from outside of the study area. They were trained on the research aim, data collection tools, method of data collection, research ethics, and on how to use ODK.

Data collection has employed record reviews, patient interviews, and anthropometric measurements. Both patient cards and service registers were reviewed following an interview. All patients were interviewed after taking their consent of participation.

## Study variables and measurements

The main outcome variable was underweight in adult HIV patients. It was defined as a Body Mass Index (BMI) of less than 18.5 kg/m$^2$. BMI is defined as the weight in kilograms divided by the square of the height in meters (kg/m2). The explanatory variables were socio-demographic and economic characteristics, clinical characteristics (presence of opportunistic infections, WHO T staging, current CD4 count, current functional status, and drug adherence), dietary diversity, substance abuse (alcohol consumption, cigarette smoking, and khat chewing), and dietary counseling.

**WHO T staging.** WHO case definitions of HIV for surveillance and revised clinical staging 1 to 4 and immunological classification of HIV-related disease in adults [27].

**Drug adherence.** Medication adherence is defined by the World Health Organization as 'the degree to which the person's behavior corresponds with the agreed recommendations from a health care provider' [28]. In the present paper, drug adherence was measured by recording the total number of pills taken by patients in one full month recall period and was assessed based on the patient's report. The adherence level was determined by dividing the actual number of pills taken by patients divided by the total number of pills to be taken which was then multiplied by 100% [30]. The patients who had scored above or equal to 95% (if

missed ≤2 doses of 30 doses) were classified as having good adherence, those from 85% to 94% (if missed 2–5 doses of 30 doses) were categorized as having fair adherence, and patients less than 85% (if missed ≥6 doses of 30 doses) were classified as having poor adherence [29, 30].

**A 24 hours' dietary diversity score (DDs).**   DDS was calculated by summing the number of unique food groups consumed during last 24 hours [23]. It was determined by asking the respondents to list all the food items consumed in the previous 24 hours preceding the assessment date, starting with the first food consumed the previous morning. If a mixed dish was eaten, participants were asked about all the ingredients of the dish. Once the recall was finished, the participant was probed for food groups to ask for food that was not mentioned. Food groups considered were cereals/roots, vegetables, fruits, legumes/lentils, meat/fish/egg and milk/dairy products. If an individual eats any quantity of any food group at least once per day, was taken into count. Therefore, DDS was calculated without considering a minimum intake for the food group. Patients who consumed 3 or fewer food groups were categorized as 'low DDS', those who consumed four to five food groups were categorized as 'medium DDS', and those who consumed equal to or greater than six food groups were 'high DDs'.

**Khat chewing.**   An individual was considered a "current khat chewer" if had chewed at least one bundle of khat per week within the last 30 days and an "ever chat chewer after starting ART" is defined if the patient had chewed at least one bundle of khat per week since starting ART [24].

**Alcohol consumption.**   Alcohol drinking is assessed using the question 'are you a current alcohol user?'; with answer 'yes' or 'no' [31].

**Cigarette smoking.**   The act of inhaling and exhaling the fumes of burning plant material. The act is most commonly associated with tobacco as smoked in a cigarette, cigar, or pipe [32]. Cigarette smoking is assessed using the question 'are you current tobacco smoker?'; with answer 'yes' or 'no' [31].

**Eating problem.**   Any difficulty of eating or swallowing food which could be secondary to loss of appetite, oral or esophageal candidiasis, nausea, vomiting, oral hairy leukoplakia etc.

*Support givers*. Are people who give home based care and support for HIV/AIDS patients that included follow-up of adherence to treatments, diet uptake and occurrence of any side effects.

**Nutrition counseling.**   The person, as identified during the nutrition assessment by asymptomatic counseling focus on (healthy eating, meal planning, achieving optimal intake of macro- and micronutrients; promoting physical activity and exercise, food and water safety) and symptomatic phase focus on (support and advice for symptom management, addressing any nutrition related complications, preventing weight loss and potential wasting, referring moderately and severely undernourished) [25].

**Patient attitude towards underweight.**   Was measured using seven items with Likert scales of measurements that ranged from "1 = strongly disagree" to "5 = Strongly agree". The items are 'Being underweight will make me vulnerable to advanced AIDS stage'; 'Being underweight will make me vulnerable to opportunistic infections (OI)'; 'Being underweight will make me a low socioeconomic engagement'; 'Being underweight will make me frequently sick'; 'Ongoing fasting will make me underweight'; 'Ongoing fasting will make me healthy'; and 'Ongoing omit of milk and animal products from main dish make me underweight'. Patients who had scored above the median value were categorized as having a positive or favorable attitude (1), and otherwise negative or unfavorable attitude (0). The Cronbach's alpha value of internal consistency or reliability was 0.93.

**Patient attitude toward care quality.**   Was measured using six items with Likert scales of measurements that ranged from "1 = strongly disagree" to "5 = Strongly agree". The items are 'ART service provider keeps my dignity and respect'; 'ART service provider gives me adequate

information'; 'ART service provider met my preferred conditions in ART services'; 'ART providers ask about my feeling since start or initiation of ART'; 'ART service provider asks me for any side effect and takes care of me'; and 'ART service provider counsels me on the importance of adherence'. Patients who had scored above the median value were categorized as having perceived good quality of care (1), and otherwise perceived poor quality of care (0). The Cronbach's alpha value of internal consistency or reliability was 0.97.

## Data quality control

The scale was calibrated to zero for every five measurements of weight. A pretest was conducted on 5% of the sample size (5 cases and 10 control) in Bishoftu military hospital, and modifications were made as appropriate. The overall data collection process was monitored on a daily basis, and feedback was given to data collectors.

## Data processing and analysis

Data were exported to SPSS version 25 for analysis from Open Data Kit (ODK) server. Data values were checked for completeness and consistency through running frequencies. Descriptive statistics were done and the characteristics of the study participants and outcome variable were presented using the mean for continuous data, and proportion for all categorical variables.

Bivariate binary logistic regression was run to select candidate variables for multivariable logistic regression. Those variables with p-value<0.2 were entered into multivariable regression analysis. In the multivariable logistic regression, p-value<0.05 was used to declare the presence of statistically significant association between underweight and explanatory variables, and 95% confidence interval was used to show the precision of measure of the strength of association. Crude odds ratios (CORs) were used to explain the strength of association between factors and dependent variables at $p<0.2$ for bivariate logistics, and adjusted odds ratios (AORs) were used to describe the strength of association between the factors and dependent variables in multivariable logistics at $p<0.05$. The fitness of the final model was assessed by the Hosmer and Lemeshow goodness-of-fit test (p = 0.44) and the variance inflation factors (VIF<10) indicated no meaningful multicollinearity between variables in the multivariable models.

## Ethical consideration

Ethical clearance was secured from the Department of Public Health Research and Ethics Committee of Adama Hospital Medical College (Ref.no.0502/K-373/14, date March 2022). Support letter was obtained from Oromia Health Bureau, Bishoftu town heath office, and Bishoftu General Hospital. An informed written consent was taken from patients prior to interviews, and permission was obtained from the health institution for patient chart reviews. Patients were informed about the aim, benefits, and risks of participating in the research, and the right to withdraw from the study at any point, without any consequence to them. Unique codes were used to identify patients, and any personal identifier was not recorded. Data were kept confidential and used only for research purposes, and not shared with a third person.

## Results

### Socio-demographic characteristics of respondents

A total of 93 cases and 186 controls were participated in the study with a response rate of 91.2% for both cases and controls. The mean age of patients was 38.6 years (SD 9.4) for cases

and 40.6 years for controls (SD 11.3) with a bit larger than one-thirds of them were in the age range of 30–39 years. About half (49.5%) of cases and 66.7% of controls were female. Over half of cases and controls were married and were orthodox religious followers. The majority of cases (69.9%) and controls (76.3%) were urban residents, and had a family size of three or fewer (66.7% for cases and 59.7% for controls) (Table 1).

## Clinical and immunological characteristics of respondents

Six out of ten cases (61.3%) and 46.2% of controls had a current CD4 count of 200–350 cells/ μl. Greater than half of cases had current WHO T-stage one (53.8%), and the majority had

**Table 1. Socio-demographic characteristics of HIV/AIDS patients on follow-up care in Bishoftu general hospital, central Ethiopia, April to May 2022.**

| Variables | Cases (n,%) | Controls(n,%) |
|---|---|---|
| Observations | 93 | 186 |
| Gender | | |
| Female | 46(49.5) | 124(66.7) |
| Male | 47(50.5) | 62(33.3) |
| Age | | |
| 20–29 | 18(19.4) | 30(16.1) |
| 30–39 | 32(34.4) | 64(34.4) |
| 40–49 | 29(31.2) | 63(33.9) |
| 50+ | 14(15.0) | 29(15.6) |
| Marital status | | |
| Married | 52(55.9) | 107(57.5) |
| Single | 8(8.6) | 17(9.1) |
| Divorced | 11(11.8) | 20(10.8) |
| Widowed | 22(23.7) | 42(22.6) |
| Religion | | |
| Orthodox | 51(54.8) | 105(56.5) |
| Muslim | 23(24.7) | 48(25.8) |
| Protestant | 19(20.5) | 33(17.7) |
| Level of education | | |
| No education | 21(22.6) | 35(18.8) |
| Primary | 32(34.4) | 72(38.7) |
| Secondary | 29(31.2) | 56(30.1) |
| Higher | 11(11.8) | 23(12.4) |
| Occupation | | |
| Monthly salaried | 34(36.6) | 63(33.9) |
| Other | 59(63.4) | 123(66.1) |
| Residence | | |
| Urban | 65(69.9) | 142(76.3) |
| Rural | 28(30.1) | 44(23.7) |
| Family size | | |
| ≤3 | 62(66.7) | 111(59.7) |
| >3 | 31(33.3) | 75(40.3) |
| Monthly income in ETB | | |
| ≤2000 | 67(72.1) | 46(24.7) |
| >2000 | 26(27.9) | 140(75.3) |

current working functional status (80.6%) and good adherence (68.8%). While, the majority of controls had current WHO T-stage one (72%), current working functional status (86.6%), and good treatment adherence (95.7%) (Table 2).

## Nutritional status, and behavioral characteristics

Six out of ten cases (58 or 62.4%) and less than half of controls (86 or 46.2%) had medium dietary diversity score (DDS) by consuming four to five food groups during the last 24 hours. One-fifth of cases (19 or 20.4%) and 33(17.7%) controls had consumed 3 or fewer food groups, and 16(17.2%) cases and 67(36.1%) controls have consumed six and more groups of food. On average cases has consumed four food groups (SD 1.2) and controls consumed five food groups (SD 1.4) (Fig 1).

Over half of the cases (53.8%) and controls (57.5%) had received dietary counseling on healthy eating, meal planning, and a nutritious diet. While less than half of the cases received counseling on optimal intake of nutrients (33.3%), water and food safety (47.3%), physical activity and exercise (26.9%), substance use (43%), and strategies for weight gain and prevention of anemia (34.4%). Overall, only a few cases and controls had received comprehensive components of counseling that were recommended in the asymptomatic (2.2% for cases and 15.1% for controls) and symptomatic phases (9.7% for cases and 20.4% for controls) (Table 3).

Over one-fifth of cases had ever consumed alcohol (22.6%) and chewed khat (21.5%). While less than one in ten cases (4.3%) and controls (5.4%) had ever smoke cigarette (Fig 2).

## Patient attitude on underweight and care quality

Two-thirds of the cases and most of the controls believed that being underweight will make them vulnerable to an advanced AIDS stage (67.7% for cases and 97.8% for controls) and opportunistic infections (65.6% for cases and 96.2% for controls). Likewise, two-thirds of cases and most of the controls believed that being underweight will cause low socioeconomic engagement and frequent sickness. Overall six out of ten cases (60.2%) and two-thirds of controls (67.2%) had a favorable attitude towards underweight.

Greater than two-thirds of cases and most of the controls agreed that providers keep their dignity and respect (71% vs 97.8%), give adequate information (67.7% vs 98.4%), met their preferred conditions in ART services (71% vs 98.9%), and provided counseling on the importance of adherence (71% vs 98.4%). Overall, 61.3% of cases and 94.1% of controls believed that the care they had received had a good quality (Table 4).

## Predictors of underweight

In the bivariate binary logistic regression, the variables including sex, current address, monthly income, presence of support giver, current CD4 count, current WHO T-staging, opportunistic infection, anemia, eating problem, dietary diversity score, counseling on optimal uptake of Macro- and micronutrients, counseling on symptom management, addressing any nutrition related complications, and khat chewing were identified as candidates to multivariable logistic regression with p-value<0.2.

In the multivariable logistic regression monthly income, presence of support giver, eating problems, dietary counseling, dietary diversity score, and khat chewing have shown statistically significant associations with underweight with p-value<0.05. The interpretation was provided below.

The odds of underweight was nearly seven times more likely among patients whose income was ≤2000 ETB as compared to patient with an income of >2000 ETB per month. Patients who had no support givers were three times more likely to be underweight compared to those

**Table 2. Clinical and immunological characteristics HIV/AIDS patients on follow-up care in Bishoftu general hospital, central Ethiopia, April to May 2022.**

| Variables | Cases (n, %) | Controls (n, %) |
|---|---|---|
| Observations | 93 | 186 |
| Baseline CD4 count | | |
| < 200 cells/μl | 37(39.8) | 60(32.3) |
| 200–350 cells/μl | 45(48.4) | 107(57.5) |
| > 350 cells/μl | 11(11.8) | 19(10.2) |
| Current CD4 count | | |
| < 200 cells/μl | 18(19.4) | 20(10.8) |
| 200–350 cells/μl | 57(61.3) | 86(46.2) |
| > 350 cells/μl | 18(19.3) | 80(43.0) |
| Baseline BMI | | |
| <18.5 kg/m$^2$ | 32(34.4) | 57(30.6) |
| 18.5–24.9 kg/m$^2$ | 61(65.6) | 129(69.4) |
| Baseline WHO T-stage | | |
| Stage I | 29(31.2) | 81(43.5) |
| Stage II | 35(37.6) | 65(34.9) |
| Stage III | 28(30.1) | 37(19.9) |
| Stage IV | 1(1.1) | 3(1.7) |
| Current WHO T-stage | | |
| Stage I | 50(53.8) | 134(72.0) |
| Stage II | 23(24.7) | 38(20.4) |
| Stage III | 20(21.5) | 14(7.6) |
| Baseline functional status | | |
| Working | 77(82.8) | 161(86.6) |
| Ambulatory | 13(14.0) | 19(10.2) |
| Bedridden | 3(3.1) | 6(3.2) |
| Current functional status | | |
| Working | 75(80.6) | 148(79.6) |
| Ambulatory | 18(19.4) | 38(20.4) |
| Adherence | | |
| Good | 64(68.8) | 178(95.7) |
| Fair | 13(14.0) | 3(1.6) |
| Poor | 16(17.2) | 5(2.7) |
| Take therapeutic Food | | |
| No | 76(81.7) | 156(83.9) |
| Yes | 17(18.3) | 30(16.1) |
| Eating problem | | |
| No | 45(48.4) | 178(95.7) |
| Yes | 48(51.6) | 8(4.3) |
| Opportunistic infection | | |
| No | 67(72.0) | 181(97.3) |
| Yes | 26(28.0) | 5(2.7) |
| Anemia | | |
| No | 70(75.3) | 179(96.2) |
| Yes | 23(24.7) | 7(3.8) |
| Follow-up intervals | | |
| ≤ 3 months | 4(4.3) | 6(3.2) |
| >3 months | 89(95.7) | 180(96.8) |

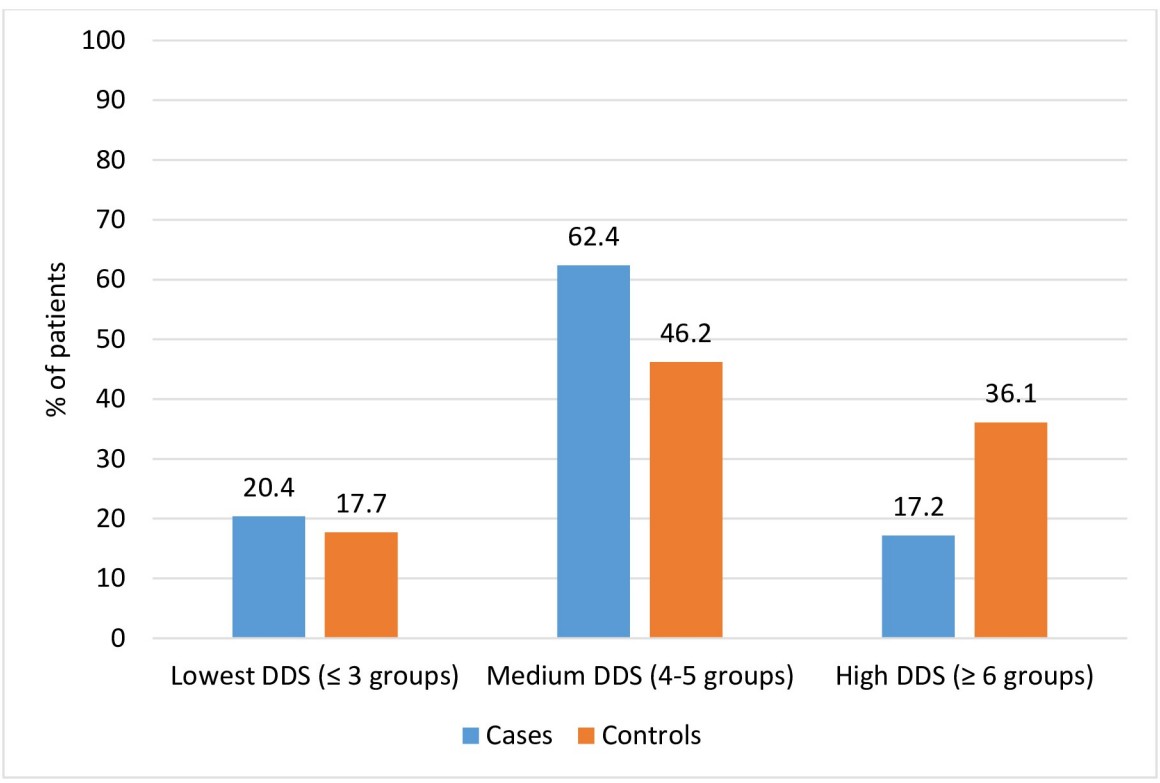

**Fig 1. Dietary diversity score (DDS) of HIV/AIDS patients on follow-up care in Bishoftu general hospital, central Ethiopia, April to May 2022.**

who had, and patients with a medium dietary diversity score (DDS of four to five) were two times more likely to be underweight compared to those with DDS score greater than or equal to six.

The odds of underweight was fourteen times more likely in patients who had eating problems compared to their counter parts and the odds underweight was 2.5 times more likely in patients who didn't receive counseling on symptom management as compared to those who received counseling. Moreover, patients who currently chew khat were three times more likely be underweight compared to those who didn't chew khat (Table 5).

## Discussion

There is a complex interaction between nutrition and HIV/AIDS. HIV progressively weakens the immune system and malnutrition increases the susceptibility to infection and death. The study revealed that monthly income, absence of support giver, eating problems, dietary diversity, khat chewing, and dietary counseling were determinants of being underweight among people living with HIV and enrolled in ART care and treatments.

The monthly income of patients was associated with being underweight among patients on HIV treatment. The odds of being underweight was nearly seven times more likely in patients whose monthly income was ≤2000 Ethiopian birr (<US $ 38) as compared to those patients whose income was greater than 2000 Ethiopian birr. This finding is comparable with a study conducted in Northwest Ethiopia, Arsi zone in the Oromia region of Ethiopia, and Mekele city in North Ethiopia indicating that PLWHIV who had a less income have the

**Table 3. Dietary counseling provided to HIV/AIDS patients on follow-up care in Bishoftu general hospital, central Ethiopia, April to May 2022.**

| Dietary counseling on- | Cases | Control |
|---|---|---|
| | n(%) | n(%) |
| Observations | 93 | 186 |
| Healthy eating, meal planning, and a nutritious diet | | |
| No | 43(46.2) | 79(42.5) |
| Yes | 50(53.8) | 107(57.5) |
| Optimal intake of macro- and micronutrients | | |
| No | 62(66.7) | 86(46.2) |
| Yes | 31(33.3) | 100(53.8) |
| HIV patients and their families on water and food safety | | |
| No | 49(52.7) | 86(46.2) |
| Yes | 44(47.3) | 100(53.8) |
| Promoting physical activity and exercise | | |
| No | 68(73.1) | 133(71.5) |
| Yes | 25(26.9) | 53(28.5) |
| Identifying and addressing related issues, such as smoking, alcohol and illicit drugs | | |
| No | 53(57.0) | 103(55.4) |
| Yes | 40(43.0) | 83(44.6) |
| Ways of achieving adequate weight gain and preventing anemia during pregnancy | | |
| No | 61(65.6) | 118(63.4) |
| Yes | 32(34.4) | 68(36.6) |
| Comprehensive dietary counseling during asymptomatic phase[a] | | |
| No | 91(97.8) | 158(84.9) |
| Yes | 2(2.2) | 28(15.1) |
| Support and advice for symptom management, addressing any nutrition related complications | | |
| No | 66(71.0) | 99(53.2) |
| Yes | 27(29.0) | 87(46.8) |
| Preventing weight loss and potential wasting | | |
| No | 49(52.7) | 94(50.5) |
| Yes | 44(47.3) | 92(49.5) |
| Referring moderately and severely undernourished adolescents and adults to specific program | | |
| No | 64(68.8) | 123(66.1) |
| Yes | 29(31.2) | 63(33.9) |
| Absorption of antiretroviral drugs | | |
| No | 49(52.7) | 97(52.2) |
| Yes | 44(47.3) | 89(47.8) |
| Managing side effects of antiretroviral therapy | | |
| No | 45(48.4) | 89(47.8) |
| Yes | 48(51.6) | 97(52.2) |
| Comprehensive dietary counseling during the symptomatic phase[b] | | |
| No | 84(90.3) | 148(79.6) |
| Yes | 9(9.7) | 38(20.4) |

[a]Counseling on all recommended components in the asymptomatic phase

[b]Counseling on all recommended components in the symptomatic phase

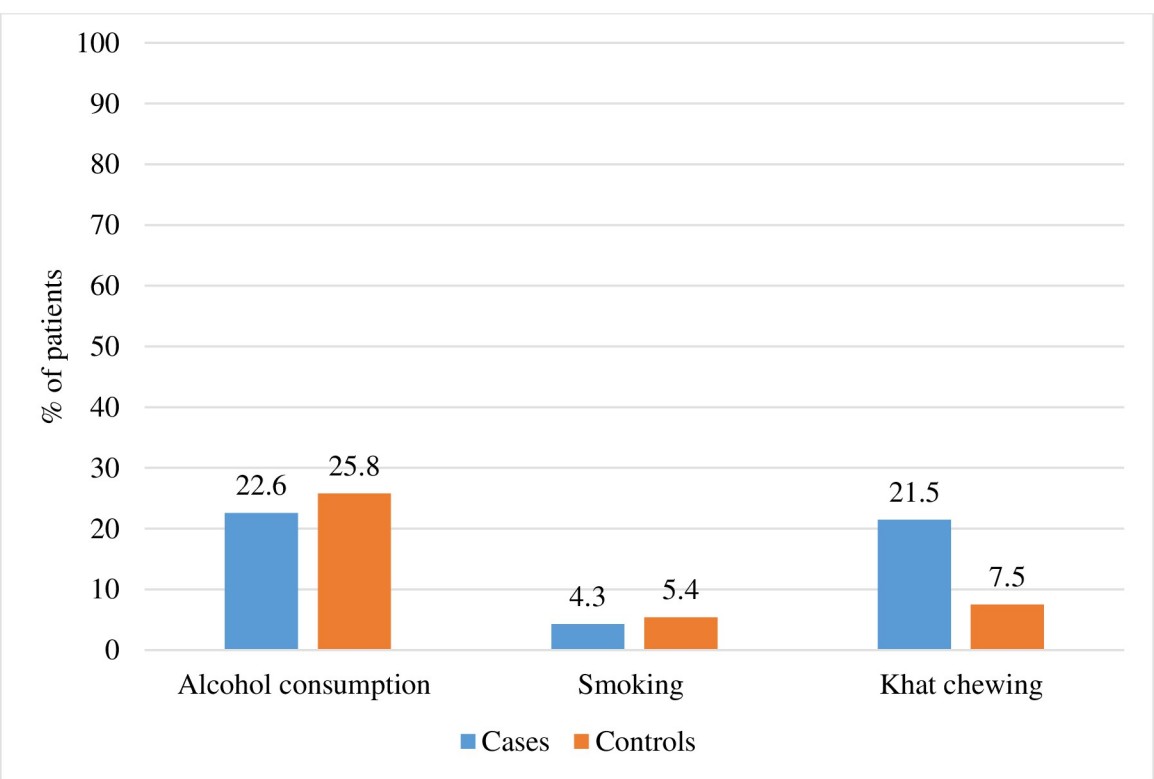

**Fig 2. Substance use among HIV/AIDS patients on follow-up care in Bishoftu general hospital, central Ethiopia, April to May 2022.**

highest odds of being undernourished [17, 33, 34]. This might be because those patients with a low income had no access to sufficient food to meet their dietary needs for productive and healthy life exposing them to deficiency of macro and micronutrients. A study conducted in different parts of Ethiopia [35–37] and in Kenya [38] and Rwanda [39] indicated that low average monthly income was associated with a low dietary diversity among HIV/AIDS patients. In contradict to our finding, that of a study done in Dilla University Referral Hospital in South Ethiopia indicated that moderate poor economic condition is not associated with underweight [40].

The HIV/AIDS care and treatment guidelines recommends that patients should have caregivers or support givers for effective uptake of treatments [16]. In the study area, 57% of cases and 83.9% of controls had support givers. The patients who had no support giver were three times more likely to be underweight compared to those patients who had people that provide them a home based care or supports.

Eating problem causes underweight on patients who are on treatment. In the present study the odds of becoming underweight while on ART were fourteen times more likely in patients who had eating problems compared to their counterparts. This finding was comparable to studies done in Northwest and Southwest Ethiopia [12, 41, 42]. The causes of eating problems are loss of appetite or anorexia, nausea, oral thrush, constipation, and bloating or heartburn [16]. HIV burns calories fast because the immune system is working hard or it may be because HIV has affected the hormones that control body metabolism and need more calories to keep normal body weight. HIV can take away appetite due to medication side effects like nausea, changes in taste or mouth tingling, symptoms of opportunistic infections and painful throat [43, 44].

**Table 4. Patient attitude towards underweight and care quality in Bishoftu general hospital, central Ethiopia, April to May 2022.**

| Variables | Cases (n, %) | Controls (n,%) |
|---|---|---|
| Observations | 93 | 186 |
| Being underweight will make me vulnerable to advanced AIDS stage | | |
| Disagreed | 30(32.3) | 4(2.2) |
| Agreed | 63(67.7) | 182(97.8) |
| Being underweight will make me vulnerable to OI | | |
| Disagreed | 32(34.4) | 7(3.8) |
| Agreed | 61(65.6) | 179(96.2) |
| Being underweight will make me low socioeconomic engagement | | |
| Disagreed | 29(31.2) | 3(1.6) |
| Agreed | 64(68.8) | 183(98.4) |
| Being underweight will make me frequently sick | | |
| Disagreed | 29(31.2) | 8(4.3) |
| Agreed | 64(68.8) | 178(95.7) |
| Ongoing fasting will make me underweight | | |
| Disagreed | 32(34.4) | 14(7.5) |
| Agreed | 61(65.6) | 172(92.5) |
| Ongoing fasting will make me healthy | | |
| Disagreed | 66(71.0) | 157(84.4) |
| Agreed | 27(29.0) | 29(15.6) |
| Ongoing omit of milk and animal products from main dish make me underweight | | |
| Disagreed | 28(30.1) | 10(5.4) |
| Agreed | 65(69.9) | 176(94.6) |
| Attitude on underweight[a] | | |
| Unfavorable (Negative) | 37(39.8) | 61(32.8) |
| Favorable (Positive) | 56(60.2) | 125(67.2) |
| ART service provider keep my dignity and respect | | |
| Disagreed | 27(29.0) | 4(2.2) |
| Agreed | 66(71.0) | 182(97.8) |
| ART service provider give me adequate information | | |
| Disagreed | 30(32.3) | 3(1.6) |
| Agreed | 63(67.7) | 183(98.4) |
| ART service provider met my preferred conditions in ART services | | |
| Disagreed | 27(29.0) | 2(1.1) |
| Agreed | 66(71.0) | 184(98.9) |
| ART providers ask about my feeling since start or initiation of ART | | |
| Disagreed | 29(31.2) | 4(2.2) |
| Agreed | 64(68.8) | 182(97.8) |
| ART service provider ask me for any side effect and take care of me | | |
| Disagreed | 29(31.2) | 2(1.1) |
| Agreed | 64(68.8) | 184(98.9) |
| ART service provider counsel me on the importance of adherence | | |
| Disagreed | 27(29.0) | 3(1.6) |
| Agreed | 66(71.0) | 183(98.4) |
| Attitude on care quality[b] | | |
| Poor quality | 36(38.7) | 11(5.9) |

(*Continued*)

**Table 4.** (Continued)

| Variables | Cases (n, %) | Controls (n,%) |
|---|---|---|
| Good quality | 57(61.3) | 175(94.1) |

Abbreviations: AIDS, acquired immune deficiency syndrome; OI, opportunistic infections; ART, antiretroviral therapy

[a]Patients who agreed on all components were categorized as 'favorable attitude' and otherwise 'unfavorable attitude'

[b]Patients who agreed on all components were categorized as 'good quality' and otherwise 'poor quality'

Low dietary diversity causes a nutritional problem in HIV-positive adults. In the present study, patients with a dietary diversity score of four to five were two times more likely underweight compared to those with dietary diversity score greater than or equals to six. This is similar to findings from studies done in Metema Hospital in Northwest Ethiopia, and Ambo in central Ethiopia where underweight was 2.5 times more likely in patients with a dietary diversity score of four to five as compared to those with a score greater than or equals six [36, 37]. Likewise, that of studies conducted in Jimma also reported that being underweight was more likely in patients with a low dietary diversity scores [20, 45]. In the current study no difference was observed in terms of underweight between patients with a lower diversity score and that of a higher dietary diversity score.

A diverse and balanced diet, rich in macro- and micronutrients, plays an important role in maintaining a healthy lifestyle and body [16]. A nutritious diet can help maintain the proper functioning of the immune system and provide the energy, protein, and micronutrients needed during all stages of HIV infection. The dietary management of HIV- and AIDS-related symptoms can prevent underweight and improve the overall health and nutritional status of PLHIVs [46]. Good food and dietary practices can decrease the effects of AIDS-related symptoms on food intake and nutrient absorption, improve comfort while eating, and prevent dehydration due to diarrhea and fever. In addition, it can help to maintain body weight, provide nutrients to compensate for losses, and strengthen the immune system [47].

Chewing khat is one of the substance abuse strongly related with developing an underweight while on treatment. In the study area the odds of underweight was three times more likely in patients who chew khat as compared to those who didn't chew khat. This evidence is supported by studies done in different parts of Ethiopia [18, 19]. This might be because khat leafs contain cathinone's, an active brain stimulant that is similar in structure and pharmacological activity to amphetamines causing decreased appetite [48].

The absence of dietary counseling aimed at symptom management was one of the determinant factors of being underweight in the study area. This finding is comparable with studies done in North and South Ethiopia [33, 49]. Early attention to the problems of underweight in patients with PLHIV has paramount importance because the timing of underweight in these patients may be more closely related to the degree of body cell mass depletion than to any specific underlying infection. Counseling, and prompt intervention can minimize wasting and helps to replete body cell mass.

The study has explored various determinants of being underweight in HIV/AIDS patients using a standard tool. However, the study was not without limitations. Recall bias and social desirability bias are potential limitations that might have affected the accuracy of information, especially related to substance use, provider fear on nutritional counseling, and 24 hours' dietary recall. Some individuals might not accurately know their age and income. In addition, data on dietary intake may be affected by seasonal variation. The study is conducted only in

**Table 5. Determinants of underweight among HIV/AIDS patients on follow-up care in Bishoftu general hospital, central Ethiopia, April to May 2022.**

| Variables | Cases (n,%) | Control (n,%) | COR (95% CI) | AOR(95% CI) |
|---|---|---|---|---|
| Observations | 93 | 186 | | |
| Sex | | | | |
| Female | 46(49.5) | 124(66.7) | 0.49(0.29–0.81)* | 0.62 (0.29–1.31) |
| Male | 47(50.5) | 62(33.3) | Ref | Ref |
| Current address | | | | |
| Within catchment | 57(61.3) | 143(76.9) | 0.48(0.28–0.82)* | 1.09(0.49–2.44) |
| Out of catchment | 36(38.7) | 43(23.1) | Ref | Ref |
| Monthly Income in ETB | | | | |
| ≤2000 | 67(72.1) | 46(24.7) | 8.08(3.65–17.89)* | 6.63(2.96–14.85)** |
| >2000 | 26(27.9) | 140(75.3) | Ref | Ref |
| Support giver | | | | |
| No | 40(43.0) | 30 (16.1) | 3.93(2.23–6.92)* | 3.22(1.38–7.50)** |
| Yes | 53(57) | 156 (83.9) | Ref | Ref |
| Current CD4 count | | | | |
| < 200 cells/μl | 18(19.4) | 20(10.8) | 4.00(1.77–9.05)* | 2.53 (0.74–8.62) |
| 200–350 cells/μl | 57(61.3) | 86(46.2) | 2.95(1.60–5.43)* | 1.17(0.44–3.10) |
| > 350 cells/μl | 18(19.4) | 80(43) | Ref | Ref |
| Current WHO T-staging | | | | |
| Stage I | 50(3.8) | 134(72) | 0.26(0.12–0.56)* | 0.49(0.15–1.59) |
| Stage II | 23(24.7) | 38(20.4) | 0.42(0.18–0.99)* | 0.61(0.18–2.12) |
| Stage III | 20(21.5) | 14(7.5) | Ref | Ref |
| Opportunistic infection | | | | |
| No | 67(72) | 181(97.3) | 0.07(0.030.19)* | 1.03(0.22–4.84) |
| Yes | 26(28) | 5(2.7) | Ref | Ref |
| Anemia | | | | |
| No | 70(75.3) | 179(96.2) | 0.12(0.05–0.29)* | 0.50(0.12–2.13) |
| Yes | 23(24.7) | 7(3.8) | Ref | Ref |
| Eating problem | | | | |
| No | 45(48.4) | 178(95.7) | Ref | Ref |
| Yes | 48(51.6) | 8(4.3) | 23.73(10.49,53.71)* | 14.48(5.06–41.40)** |
| Dietary diversity score | | | | |
| Low(≤ 3) | 19(20.4) | 33(17.7) | 2.41(1.10–5.29)* | 1.47(0.46–4.70) |
| Medium (4–5) | 58(62.4) | 86(46.2) | 2.82(1.49–5.35)* | 2.36(1.92–6.08)** |
| High (≥6) | 16(17.2) | 67(36.1) | Ref | Ref |
| Counseling on optimal uptake of Macro- and micronutrients | | | | |
| No | 62(66.7) | 86(46.2) | 2.33(1.38–3.91)* | 1.18(0.55–2.54) |
| Yes | 31(33.3) | 100(53.8) | Ref | Ref |
| Counseling on symptom management, addressing any nutrition related complications | | | | |
| No | 66(71.0) | 99(53.2) | 2.15(1.26–3.66)* | 2.53(1.11–5.72)** |
| Yes | 27(29.0) | 87(46.8) | Ref | Ref |
| Khat chewing | | | | |
| No | 73(78.5) | 172(92.5) | Ref | Ref |
| Yes | 20(21.5) | 14(7.5) | 3.37(1.61,7.02)* | 3.36(1.99,11.33)** |

* p-value ≤0.2,

** p< 0.05

Abbreviations: COR, crude odds ratio; AOR, adjusted odds ratio; CI, confidence interval; WHO, world health organization

one heath facility, and may not show the variations in the different settings. Thus, the findings should be used with caution.

## Conclusions

The study has concluded that monthly income, absence of support giver, eating problems, dietary diversity, substance use (khat chewing) and the absence of dietary counseling focused on symptoms management were determinants of underweight among HIV/AIDS patients on treatments. Increasing patient income, creating awareness on dietary diversity and food uptake, ensuring access to support giver, and educating on substance abuse has paramount importance to improve their health. Therefore, strategies and programs that targets income generating activities and awareness creations intervention focused on nutrition and substance abuse should be strengthened. Strategies and programs that targets mental health with substance abuse to PLWHIV should be integrated to the care and treatment program. Moreover, early identification of nutritional status and eating problems with a prompt action should be enhanced in the ART clinics.

## Supporting information

**S1 Data.**
(XLSX)

## Acknowledgments

The researchers thank the research participants and administrative structure in the different hierarchies of the study setting for their due cooperation and involvement.

## Author Contributions

**Conceptualization:** Mesfin Gashaw Assefa, Alem Deksisa, Mariama Abdo, Obse Tamiru Alemayehu, Dawit Wolde Daka.

**Data curation:** Mesfin Gashaw Assefa, Alem Deksisa, Mariama Abdo, Obse Tamiru Alemayehu, Dawit Wolde Daka.

**Formal analysis:** Mesfin Gashaw Assefa, Alem Deksisa, Mariama Abdo, Obse Tamiru Alemayehu, Dawit Wolde Daka.

**Investigation:** Mesfin Gashaw Assefa, Alem Deksisa, Mariama Abdo, Obse Tamiru Alemayehu, Dawit Wolde Daka.

**Methodology:** Mesfin Gashaw Assefa, Alem Deksisa, Mariama Abdo, Obse Tamiru Alemayehu, Dawit Wolde Daka.

**Project administration:** Alem Deksisa, Mariama Abdo, Dawit Wolde Daka.

**Software:** Mesfin Gashaw Assefa, Alem Deksisa, Mariama Abdo, Obse Tamiru Alemayehu, Dawit Wolde Daka.

**Supervision:** Mesfin Gashaw Assefa, Alem Deksisa, Mariama Abdo, Obse Tamiru Alemayehu, Dawit Wolde Daka.

**Validation:** Mesfin Gashaw Assefa, Alem Deksisa, Mariama Abdo, Dawit Wolde Daka.

**Visualization:** Mesfin Gashaw Assefa, Alem Deksisa, Dawit Wolde Daka.

**Writing – original draft:** Dawit Wolde Daka.

**Writing – review & editing:** Mesfin Gashaw Assefa, Alem Deksisa, Mariama Abdo, Obse
Tamiru Alemayehu, Dawit Wolde Daka.

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
