## [Decision Letter · Decision Letter 0]

13 Feb 2023

PONE-D-22-26786Predictors of Underweight Among Adult Patients Receiving Antiretroviral Therapy in Bishoftu General Hospital, Central Ethiopia: Case-Control StudyPLOS ONE

Dear Dr. Daka,

Thank you for submitting your manuscript to PLOS ONE. After careful consideration, we feel that it has merit but does not fully meet PLOS ONE’s publication criteria as it currently stands. Therefore, we invite you to submit a revised version of the manuscript that addresses the points raised during the review process.

We look forward to receiving your revised manuscript.

Kind regards,

Palash Chandra Banik, MPhil

Academic Editor

PLOS ONE

Journal Requirements:

Reviewers' comments:

Reviewer's Responses to Questions

**Comments to the Author**

1. Is the manuscript technically sound, and do the data support the conclusions?

Reviewer #1: Yes

Reviewer #2: Yes

Reviewer #3: Yes

2. Has the statistical analysis been performed appropriately and rigorously? 

Reviewer #1: Yes

Reviewer #2: Yes

Reviewer #3: Yes

3. Have the authors made all data underlying the findings in their manuscript fully available?

Reviewer #1: No

Reviewer #2: Yes

Reviewer #3: Yes

4. Is the manuscript presented in an intelligible fashion and written in standard English?

Reviewer #1: Yes

Reviewer #2: Yes

Reviewer #3: Yes

5. Review Comments to the Author

Reviewer #1: � It is necessary to reflect on the sampling technique used to select the study participants. I advised the authors of this manuscript to consider sampling techniques for case-control studies i.e. base or case-base sampling, cumulative density or survivor sampling, and incidence density or risk set sampling. I think in this case it is a cumulative density or survivor sampling.

Cite the appropriate references for the operational definitions. I think references for the following are available…The DDS was calculated as the sum of the food groups consumed over 24 hours(reference). WHO staging: WHO case definitions of HIV for surveillance and revised clinical staging 1 to 4 and immunological classification of HIV-related disease in adults (reference). Drug Adherence: Medication adherence is defined by the World Health Organization as ‘the degree to which the person's behavior corresponds with the agreed recommendations from a health care provider (put the reference here)

How do you measure the patient attitude towards underweight and care quality? Did the authors of this manuscript use a scale like the Likert scale, if so how do you measure internal consistency/, How the internal consistency reliability is quantified? What is the value of Cronbach's coefficient alpha? If Crombach's alphas of the attitude scale is available, it is recommended to reflect it

Reviewer #2: Comments

1. In the case using abbreviation/acrimony in the first place use full word then abbreviation in Bereket.

Correct starting from abstract

2. In result section - Monthly income of patients <1000 birr (aOR=10.18, 95% CI: 3.31, 31.26) and 1000-

2000 birr (aOR=5.56, 95% CI: 2.33, 13.24)

Comment – aOR- Correct with AOR and you categorization of income without logical reasoning that is why AOR Very wide need adjustment.

3. In result section-Tables (2-5)title not incomplete example table 2. Clinical and immunological characteristics not incomplete should include study area and study period.

4. In result section line 290-295- In the multivariable logistic regression, variables including monthly income, presence of 292 caregiver, eating problems, dietary counselling, dietary diversity score, and khat chewing have 293 shown statistically significant associations with underweight. Though the remaining variables 294 have showed associations in the binary logistic regression with a p-value

Comment – binary include bivariate & multivariable regression analysis, based on this concept the above interpretation is wrong – the author should set criteria for those variables from bivariate analysis to multivariable analysis. i.e. p-value<0.05 is not serve as a criteria for bivariate?

Reviewer #3: Manuscript Number: PONE-D-22-26786

Article Type: Research Article

Full Title: Predictors of Underweight Among Adult Patients Receiving Antiretroviral Therapy in Bishoftu General Hospital, Central Ethiopia: Case-Control Study

Reviewer: Habtemu Jarso Hebo

Comments

# Abstract

Methods:

- Sampling technique is not mentioned.

- Type of logistic regression is not specified. Is it binary, multinomial, or ordinal?

- Why both 95% CI and p-value < 0.05 to declare statistical significance? P-value is sufficient to declare statistical significance and 95% CI is important to show the precision of measure of strength of association.

Conclusion:

- Rewrite conclusion as follows:

“This study revealed that household dietary diversity, counseling and education on nutrition, monthly income, eating problem, caregiver and chat chewing were predictors of underweight in HIV patients. This inquires an integrated nutritional intervention including income-generating activities, counseling and education on nutrition and bad habits, and regular monitoring of the nutritional status during clinic visits.”

# Introduction

- Line no. 50-52: The sentence is long and compound; it should be broken down into two as follows: “Malnutrition reduces the capacity of the body to fight HIV infection by compromising the immune system. HIV/AIDS resulted in undernutrition through opportunistic infections by causing immune dysfunction in manifold ways(1,2).”

- Line no. 59-61: the information is not relevant; it presents about general population rather than PLWH. It is also old data (2014).

- Line no. 67-70: the sentence if long and compound; it should be broken down into two as follows:

“Underweight significantly shortens the time to develop OIs in adults living with HIV compared to a well-nourished patients. The average time to develop OIs in underweight patients is estimated as 17 months and that of well-nourished is 26 months(12).”

# Methods and Materials

- Line no. 95-97: the phrase, “among adult HIV/AIDS patients who were receiving antiretroviral therapy or treatments in Bishoftu general hospital” is redundant and not important as part of the study design.

- Line no. 111-113: “420 were children less than 18 years old” – but in practice, those aged 15 and above years are treated as adults. So, you have to present those who were under-fifteen (<15 years) as children. “25 were lost-to-follow-up and 928 were drop-out or hadn’t been in contact with the clinic for more than three months”. But other authors have defined those patients who hadn’t been in contact with the clinic for more than three months as “loss to follow-up” and those who hadn’t been in contact with the clinic for less than three months as “default”. So, how do you arbitrate your definition with others? What is your reference?

- Line no. 117: In other studies, adults are defined 15 years and above.

- Line no. 118: editorial error; 18.5 is repeated after 24.9.

- Line no. 128: editorial error; edit as, “proportion of poor adherence to ART (28.11% for cases and 12.6% for controls).

- Line no. 135 & 136: typo errors: literatures and information’s; correct as, literature and information

- Line no. 154: typos error: s/he is not recommended in scientific writing.

- Line no. 156 – 163: regarding the measurement of nutrition counseling, it should be nutrition counseling during asymptomatic period. If it included symptomatic period, patients might have been given nutrition counseling very recently (just before the date of data collection) and so that they are malnourished during data collection. This underestimates the beneficial effect of nutrition counseling as patients are provided with it but still malnourished. The other thing s terminology consistency; i.e. nutrition counseling or dietary counseling?

- Line no. 182-185: explanatory variables (basic socio-demographic/economic and clinical characteristics) must be current profile rather than baseline characteristics. It’s not mentioned those characteristics are profiles on the date of interview.

- Line no. 186: WHO staging; it should be WHO T-staging because the patients were on treatment at least for 6 months.

- Line no. 188: definition for adherence is just scientific definition provided by WHO. What is operational definition of adherence in your study? When do you say a patient is adherent or not?

- Line #191-193 is redundant; it is already present under measurement section. The description in line #194-198 can be added to that section.

- Line #199: alcohol consumption: the definition is not relevant as described here. Clear operational definition should be provided like you did for khat chewing.

- Line #202: Cigarette smoking: this is also not defined. When do you say a patient is smoker or not? Precise operational definition should be provided.

- Line #204: Eating problem is better defined as, “any difficulty of eating or swallowing food which could be secondary to loss of appetite, oral or esophageal candidiasis, nausea, vomiting, oral hairy leukoplakia, etc”.

- Variables and measurement sections including operational definitions should merged to avoid redundancy. References should be provided for operational definitions.

- Line #206: from where was data exported to SPSS?

- Line #210: binary logistic regression is the general term for type of logistic regression when outcome is dichotomous. But the appropriate term for the regression containing one independent variable is bivariate or bivariable regression. So, yours is bivariate binary logistic regression.

- Data analysis: did you check multicollinearity and interactions? How and what did you find?

# Results

- Line #229: you have reported 102 cases and 204 controls in the abstract section. But this (93 cases and 186 controls) is what you have to report in the abstract section too. The response rate is exactly the same in both groups because refused controls were replaced to include two controls for each case included. But it’s expected to come across some refusals among controls selected for each case.

- Line #231: neither two-thirds of cases nor controls were in the age group 30-39, but a bit larger than one-third of both cases and controls were in this age group.

- Line #236: Bishoftu General Hospital; not only Bishoftu hospital.

- Line #237: April to May 2022; not only 2022.

- Line #240-244: it is great that you have considered the current clinical characteristics in your study. Some typos correction are WHO T-stage one; not clinical stage and current working functional status; not working current functional status.

- Line #245: The caption for Table 2 is not complete. The population, where and when components are missing.

- Line #245: baseline and current clinical characteristics could have collinearity and must be explored before proceeding to multivariable analysis.

- Line #255: the caption for figure 1 is also not complete.

- Line #263: caption is not complete for table 3.

- Line #266 and 267: ever alcohol consumption and ever smoking cigarette: these are not defined in the previous section.

- Line #269: caption for figure 2 is not complete.

- Table 4: ongoing fasting will make me underweight, agreed = 65.6% for cases and 92.5% for controls. Similarly, ongoing fasting will make me healthy, agreed = 71% for cases and 76.9% for controls. The two arguments are opposite but the results for agree are more than two-thirds in both cases. How could this happen?

- Attitude towards underweight and care quality are not mentioned in the variables section. Their measurement was also not described in measurement section.

- Line #290: how many variables were found candidates for multivariable regression and included in the model?

- Line #306: Table 5, the caption is not complete.

- Table 5: the variable caregiver is mentioned here and in the abstract. It is not described in the variables and measurement sections. This variable is commonly for pediatric patients. What does it mean for adults and how did you assess it?

- Table 5: for variable eating problem, why did you take the ‘yes’ category as a reference group while it is the category with small frequency? Is it not more logical and easy to interpret the ‘yes’ category is at higher risk of underweight compared to the ‘no’ category? The same is true for a variable ‘khat chewing’.

- Table 5: for variable dietary diversity score, how logical do you think when those having medium DDS are at higher risk of underweight compared to those with high DDS but those having low DDS are not different from those with high DDS? It is expected for those with low DDS to have higher risk than medium DDS group when compared to high DDS group. Could your result be because of measurement problem and thus misclassification bias? Suspect and mention this in the limitation section.

# Discussion

- The discussion needs revision for grammar and typos errors. Again authors should compare apples with apples and orange with orange; but not apples with orange. For instance, they compared socio-economic status measured as monthly income with the one measured as wealth index.

- Justification should be scientifically sound and supported by references.

# Conclusion

- The first sentence is not relevant.

# Diagrams

- The diagrams are not relevant as this is a case-control study.

- The second diagram is even not appropriate; the proportions don’t sum up to hundred. In such stacked bar graphs, the proportions should sum up to hundred.

6. PLOS authors have the option to publish the peer review history of their article (what does this mean?). If published, this will include your full peer review and any attached files.

Reviewer #1: No

Reviewer #2: **Yes: **Dr. ESubalew Tesfahun

Reviewer #3: **Yes: **Habtemu Jarso Hebo

---

## [Author Response · Author response to Decision Letter 0]

14 Apr 2023

April 11, 2023

To: The Editor,

Subject: Response to reviewers’ comments 

Dear Editor, thank you very much for giving us the opportunity to submit a revised version of our manuscript and we appreciate the reviewers for providing us with detailed comments and feedback on the paper. Point-by-point responses to reviewer comments are described below. The page and line numbers appended in the responses are based on the cleaned copy of the manuscript file. 

Kindly regards,

Dawit Wolde Daka

Corresponding Author

E-mail: dave86520@gmail.com

REVIEWER 1:

Reviewer: It is necessary to reflect on the sampling technique used to select the study participants. I advised the authors of this manuscript to consider sampling techniques for case-control studies i.e. base or case-base sampling, cumulative density or survivor sampling, and incidence density or risk set sampling. I think in this case it is a cumulative density or survivor sampling. 

Response: Thank you. We have described the sampling strategy in the revised manuscript (Lines # 131 to 132) 

Reviewer: Cite the appropriate references for the operational definitions. I think references for the following are available…The DDS was calculated as the sum of the food groups consumed over 24 hours(reference). WHO staging: WHO case definitions of HIV for surveillance and revised clinical staging 1 to 4 and immunological classification of HIV-related disease in adults (reference). Drug Adherence: Medication adherence is defined by the World Health Organization as ‘the degree to which the person's behavior corresponds with the agreed recommendations from a health care provider (put the reference here)

Response: Thank you. As suggested, we have added references to operational definitions in the revised manuscript (Lines # 164 and after wards). 

Reviewer: How do you measure the patient attitude towards underweight and care quality? Did the authors of this manuscript use a scale like the Likert scale, if so how do you measure internal consistency/, How the internal consistency reliability is quantified? What is the value of Cronbach's coefficient alpha? If Crombach's alphas of the attitude scale is available, it is recommended to reflect it

Response: Thank you. We have added text that describes the measurements of attitude towards underweight and care quality (Lines # 207 to 225) in the revised manuscript. The Cronbach alpha score was also included in the text description.

REVIEWER 2:

Reviewer: In the case of using abbreviation/acrimony in the first place use the full word and then the abbreviation in Bracket. Correct starting from abstract

Response: Thank you. We have corrected this in the revised manuscript. 

Reviewer: In result section - Monthly income of patients <1000 birr (aOR=10.18, 95% CI: 3.31, 31.26) and 1000- 2000 birr (aOR=5.56, 95% CI: 2.33, 13.24) Comment – aOR- Correct with AOR and you categorization of income without logical reasoning that is why AOR Very wide need adjustment

Response: Thank you. We have corrected this in the revised manuscript (Lines # 39 to 40)

Reviewer: In result section-Tables (2-5) title not incomplete example table 2. Clinical and immunological characteristics not incomplete should include study area and study period.

Response: Thank you. We have modified table titles in the revised manuscript. 

Reviewer: In result section line 290-295- In the multivariable logistic regression, variables including monthly income, presence of caregiver, eating problems, dietary counselling, dietary diversity score, and khat chewing have shown statistically significant associations with underweight. Though the remaining variables have showed associations in the binary logistic regression with a p-value Comment – binary include bivariate & multivariable regression analysis, based on this concept the above interpretation is wrong – the author should set criteria for those variables from bivariate analysis to multivariable analysis. i.e. pvalue<0.05 is not serve as a criterion for bivariate?

Response: Thank you. We have revised the text, and the information in the table 5 in the revised manuscript. Cutoff p-value for bivariate analysis was p-value<0.2 and that of multivariable regression was p-value<0.05. 

REVIEWER 3: 

Reviewer: # Abstract Methods: Sampling technique is not mentioned. - Type of logistic regression is not specified. Is it binary, multinomial, or ordinal? - Why both 95% CI and p-value < 0.05 to declare statistical significance? P-value is sufficient to declare statistical significance and 95% CI is important to show the precision of measure of strength of association. 

Response: Thank you. We have added text and modified this part in the revised manuscript (Lines # 31 to 37). 

Reviewer: Conclusion: - Rewrite conclusion as follows: “This study revealed that household dietary diversity, counseling and education on nutrition, monthly income, eating problem, caregiver and chat chewing were predictors of underweight in HIV patients. This inquires an integrated nutritional intervention including income-generating activities, counseling and education on nutrition and bad habits, and regular monitoring of the nutritional status during clinic visits.”

Response: Thank you. As suggested, we have corrected the conclusion in the revised manuscript. 

Reviewer: # Introduction - Line no. 50-52: The sentence is long and compound; it should be broken down into two as follows: “Malnutrition reduces the capacity of the body to fight HIV infection by compromising the immune system. HIV/AIDS resulted in undernutrition through opportunistic infections by causing immune dysfunction in manifold ways (1,2).”

Response: Thank you. We have modified the text as suggested. 

Reviewer: # Introduction -Line no. 59-61: the information is not relevant; it presents about general population rather than PLWH. It is also old data (2014). 

Response: Thank you. We have revised it as suggested. 

Reviewer: # Introduction -Line no. 67-70: the sentence if long and compound; it should be broken down into two as follows: “Underweight significantly shortens the time to develop OIs in adults living with HIV compared to well-nourished patients. The average time to develop OIs in underweight patients is estimated as 17 months and that of the well-nourished is 26 months (12). 

Response: Thank you. We have rewritten the sentences as suggested. 

Reviewer: # Methods and Materials-Line no. 95-97: the phrase, “among adult HIV/AIDS patients who were receiving antiretroviral therapy or treatments in Bishoftu general hospital” is redundant and not important as part of the study design.

Response: Thank you. We have modified the text as suggested. 

Reviewer: Line no. 111-113: “420 were children less than 18 years old” – but in practice, those aged 15 and above years are treated as adults. So, you have to present those who were under fifteen (<15 years) as children. “25 were lost-to-follow-up and 928 were drop-out or hadn’t been in contact with the clinic for more than three months”. But other authors have defined those patients who hadn’t been in contact with the clinic for more than three months as “loss to follow-up” and those who hadn’t been in contact with the clinic for less than three months as “default”. So, how do you arbitrate your definition with others? What is your reference?

Response: Thank you. We have revised the text and made it clearer (Lines # 109 to 112). 

Reviewer: Line no. 117: In other studies, adults are defined 15 years and above.

Response: Thank you. The study has included patients aged 15 years and older, and we have revised the description in the revised manuscript. 

Reviewer: Line no. 118: editorial error; 18.5 is repeated after 24.9.

Response: We have corrected. 

Reviewer: Line no. 128: editorial error; edit as, “proportion of poor adherence to ART (28.11% for cases and 12.6% for controls).

Response: We have corrected. 

Reviewer: Line no. 135 & 136: typo errors: literatures and information’s; correct as, literature and information

Response: We have corrected. 

Reviewer: Line no. 154: typos error: s/he is not recommended in scientific writing. 

Response: We have corrected as suggested. 

Reviewer: Line no. 156 – 163: regarding the measurement of nutrition counseling, it should be nutrition counseling during asymptomatic period. If it included symptomatic period, patients might have been given nutrition counseling very recently (just before the date of data collection) and so that they are malnourished during data collection. This underestimates the beneficial effect of nutrition counseling as patients are provided with it but still malnourished. The other thing s terminology consistency; i.e. nutrition counseling or dietary counseling?

Response: Thank you. The measurement of nutrition counseling has focused both on the asymptomatic and symptomatic phases. We aimed to assess whether patients had received counseling on those contents in order to improve their nutritional status. The guideline suggests that all patients who are on care and treatment should be counseled on the components of asymptomatic and symptomatic counseling to improve their nutritional status. We have corrected the terminology used. 

Reviewer: Line no. 182-185: explanatory variables (basic socio-demographic/economic and clinical characteristics) must be current profile rather than baseline characteristics. It’s not mentioned those characteristics are profiles on the date of the interview.

Response: Thank you. Current patient information was used in the analysis. 

Reviewer: Line no. 186: WHO staging; it should be WHO T-staging because the patients were on treatment at least for 6 months.

Response: Thank you. We have corrected it as suggested. 

Reviewer: Line no. 188: definition for adherence is just scientific definition provided by WHO. What is operational definition of adherence in your study? When do you say a patient is adherent or not?

Response: Thank you. We have corrected the definition in the revised manuscript (Lines # 166 to 175). 

Reviewer: Line #191-193 is redundant; it is already present under measurement section. The description in line #194-198 can be added to that section.

Response: We have corrected this part as suggested. 

Reviewer: Line #199: alcohol consumption: the definition is not relevant as described here. Clear operational definition should be provided like you did for khat chewing. - Line #202: Cigarette smoking: this is also not defined. When do you say a patient is smoker or not? Precise operational definition should be provided. 

Response: We have revised the operational definitions (Lines # 191 to 196). 

Reviewer: Line #204: Eating problem is better defined as, “any difficulty of eating or swallowing food which could be secondary to loss of appetite, oral or esophageal candidiasis, nausea, vomiting, oral hairy leukoplakia, etc”. - Variables and measurement sections including operational definitions should merged to avoid redundancy. References should be provided for operational definitions.

Response: We have revised this part as suggested. 

Reviewer: Line #206: from where was data exported to SPSS?

Response: Data were exported from the ODK server to SPSS. We have corrected the statement (Line # 232). 

Reviewer: Line #210: binary logistic regression is the general term for type of logistic regression when outcome is dichotomous. But the appropriate term for the regression containing one independent variable is bivariate or bivariable regression. So, yours is bivariate binary logistic regression.

Response: We have corrected in the revised manuscript.

Reviewer: Data analysis: did you check multicollinearity and interactions? How and what did you find? 

Response: Thank you. As suggested we have checked multicollinearity and the findings were presented in the revised manuscript (Lines # 245 to 246). 

Reviewer: # Results - Line #229: you have reported 102 cases and 204 controls in the abstract section. But this (93 cases and 186 controls) is what you have to report in the abstract section too. The response rate is exactly the same in both groups because refused controls were replaced to include two controls for each case included. But it’s expected to come across some refusals among controls selected for each case.

Response: We have corrected this in the revised manuscript. 

Reviewer: Line #231: neither two-thirds of cases nor controls were in the age group 30-39, but a bit larger than one-third of both cases and controls were in this age group. 

Response: We have corrected the statement. 

Reviewer: Line #236: Bishoftu General Hospital; not only Bishoftu hospital.

Response: We have corrected.

Reviewer: Line #237: April to May 2022; not only 2022.

Response: We have corrected. 

Reviewer: Line #240-244: it is great that you have considered the current clinical characteristics in your study. Some typos correction are WHO T-stage one; not clinical stage and current working functional status; not working current functional status

Response: Thank you. We have corrected the typology errors as suggested. 

Reviewer: Line #245: The caption for Table 2 is not complete. The population, where and when components are missing.

Response: Thank you. We have rewritten it. 

Reviewer: Line #245: baseline and current clinical characteristics could have collinearity and must be explored before proceeding to multivariable analysis. 

Response: Thank you. In the analysis, we have considered current clinical characteristics. We have diagnosed the presence of multicollinearity and the result is presented in the revised manuscript (Lines # 245 to 246). 

Reviewer: Line #255: the caption for figure 1 is also not complete. - Line #263: caption is not complete for table 3. Line #269: caption for figure 2 is not complete. 

Response: We have corrected. 

Reviewer: Line #266 and 267: ever alcohol consumption and ever smoking cigarette: these are not defined in the previous section.

Response: We have included the definition in the methods part (Lines # 191 to 196). 

Reviewer: Table 4: ongoing fasting will make me underweight, agreed = 65.6% for cases and 92.5% for controls. Similarly, ongoing fasting will make me healthy, agreed = 71% for cases and 76.9% for controls. The two arguments are opposite but the results for agree are more than two-thirds in both cases. How could this happen?

Response: Thank you. The previously presented result is without reverse coding and now we have corrected it. 

Reviewer: Attitude towards underweight and care quality are not mentioned in the variables section. Their measurement was also not described in measurement section.

Response: Thank you. Now we have included the definitions in the revised manuscript (Lines # 207 to 225). 

Reviewer: Line #290: how many variables were found candidates for multivariable regression and included in the model? - Line #306: Table 5, the caption is not complete.

Response: We have described the candidate variables for multivariable logistic regression in the revised manuscript (Lines # 328 to 333). We have corrected the caption. 

Reviewer: Table 5: the variable caregiver is mentioned here and in the abstract. It is not described in the variables and measurement sections. This variable is commonly for pediatric patients. What does it mean for adults and how did you assess it?

Response: The variable is intended to measure the presence of support givers rather than caregivers. We have corrected the variable and defined it in the revised manuscript (Lines # 199 to 200).

Reviewer: Table 5: for variable eating problem, why did you take the ‘yes’ category as a reference group while it is the category with small frequency? Is it not more logical and easy to interpret the ‘yes’ category is at higher risk of underweight compared to the ‘no’ category? The same is true for a variable ‘khat chewing’. - Table 5: for variable dietary diversity score, how logical do you think when those having medium DDS are at higher risk of underweight compared to those with high DDS but those having low DDS are not different from those with high DDS? It is expected for those with low DDS to have higher risk than medium DDS group when compared to high DDS group. Could your result be because of measurement problem and thus misclassification bias? Suspect and mention this in the limitation section

Response: We have corrected this. 

Reviewer: # Discussion - The discussion needs revision for grammar and typos errors. Again authors should compare apples with apples and orange with orange; but not apples with orange. For instance, they compared socio-economic status measured as monthly income with the one measured as wealth index. - Justification should be scientifically sound and supported by references.

Response: We have revised discussion as suggested. 

Reviewer: # Conclusion - The first sentence is not relevant.

Response: We have corrected as suggested. 

Reviewer: # Diagrams - The diagrams are not relevant as this is a case-control study. - The second diagram is even not appropriate; the proportions don’t sum up to hundred. In such stacked bar graphs, the proportions should sum up to hundred.

Response: Thank you for the suggestion. We have corrected figure 2.

---

## [Decision Letter · Decision Letter 1]

5 Sep 2023

Predictors of Underweight Among Adult Patients Receiving Antiretroviral Therapy in Bishoftu General Hospital, Central Ethiopia: Case-Control Study

PONE-D-22-26786R1

Dear Dr. Daka,

We’re pleased to inform you that your manuscript has been judged scientifically suitable for publication and will be formally accepted for publication once it meets all outstanding technical requirements.

Kind regards,

Steve Zimmerman, PhD

Associate Editor, PLOS ONE

Additional Editor Comments (optional):

Reviewers' comments:

Reviewer's Responses to Questions

**Comments to the Author**

1. If the authors have adequately addressed your comments raised in a previous round of review and you feel that this manuscript is now acceptable for publication, you may indicate that here to bypass the “Comments to the Author” section, enter your conflict of interest statement in the “Confidential to Editor” section, and submit your "Accept" recommendation.

Reviewer #2: All comments have been addressed

Reviewer #3: All comments have been addressed

2. Is the manuscript technically sound, and do the data support the conclusions?

Reviewer #2: Yes

Reviewer #3: Yes

3. Has the statistical analysis been performed appropriately and rigorously? 

Reviewer #2: Yes

Reviewer #3: Yes

4. Have the authors made all data underlying the findings in their manuscript fully available?

Reviewer #2: Yes

Reviewer #3: Yes

5. Is the manuscript presented in an intelligible fashion and written in standard English?

Reviewer #2: Yes

Reviewer #3: Yes

6. Review Comments to the Author

Reviewer #2: All comments given by me has been corrected. No additional comment on this version.

This research not violate research ethics when conducting research on human subjects doing this research by minimize harms and risks and maximize benefits; respect human dignity, privacy, and autonomy; take special precautions with vulnerable populations; and strive to distribute the benefits and burdens of research fairly.

Reviewer #3: (No Response)

7. PLOS authors have the option to publish the peer review history of their article (what does this mean?). If published, this will include your full peer review and any attached files.

Reviewer #2: **Yes: **Esubalew Tesfahun

Reviewer #3: **Yes: **Habtemu Jarso Hebo

---

## [Editor Report · Acceptance letter]

12 Sep 2023

PONE-D-22-26786R1 

Predictors of Underweight Among Adult Patients Receiving Antiretroviral Therapy in Bishoftu General Hospital, Central Ethiopia: Case-Control Study 

Dear Dr. Daka:

I'm pleased to inform you that your manuscript has been deemed suitable for publication in PLOS ONE. Congratulations! Your manuscript is now with our production department. 

Kind regards, 

on behalf of

Dr Steve Zimmerman 

Staff Editor

PLOS ONE